# Antisense Oligonucleotides for Rapid Translation of Gene Therapy in Glioblastoma

**DOI:** 10.3390/cancers16101944

**Published:** 2024-05-20

**Authors:** Jelisah F. Desgraves, Mynor J. Mendez Valdez, Jay Chandar, Muhammet Enes Gurses, Lisa Henderson, Jesus R. Castro, Deepa Seetheram, Michael E. Ivan, Ricardo J. Komotar, Ashish H. Shah

**Affiliations:** 1Section of Virology and Immunotherapy, Sylvester Comprehensive Cancer Center, Miller School of Medicine, University of Miami, Miami, FL 33136, USA; jfd81@med.miami.edu (J.F.D.); mynor.mendezvaldez@med.miami.edu (M.J.M.V.); jchan119@med.fiu.edu (J.C.); lisa.henderson@nih.gov (L.H.); jxc2506@miami.edu (J.R.C.); sxd1229@med.miami.edu (D.S.); ashah@med.miami.edu (A.H.S.); 2Department of Neurosurgery, Miller School of Medicine, University of Miami, 1095 NW 14th Terrace (D4-6), Miami, FL 33136, USA; mei12@miami.edu (M.E.I.); rkomotar@med.miami.edu (R.J.K.)

**Keywords:** antisense oligonucleotides, high-grade gliomas, blood–brain barrier, brain tumor, gene therapy, central nervous system, adjuvant therapy

## Abstract

**Simple Summary:**

The treatment for glioblastoma, a highly aggressive brain tumor, has not significantly improved in the past two decades. Current standard care involves maximal surgical removal of the tumor followed by chemotherapy and radiation, yet overall survival rates remain low. Gene therapy, particularly the use of antisense oligonucleotides (ASOs), is a promising new approach. ASOs can target and inhibit specific genes involved in tumor growth. This review discusses the biochemical mechanisms of ASO therapy, recent advancements in their use for glioblastoma, and their potential as an additional treatment in clinical practice.

**Abstract:**

Purpose: The limited efficacy of current treatments for malignant brain tumors necessitates novel therapeutic strategies. This study aimed to assess the potential of antisense oligonucleotides (ASOs) as adjuvant therapy for high-grade gliomas, focusing on their CNS penetration and clinical translation prospects. Methods: A comprehensive review of the existing literature was conducted to evaluate the implications of ASOs in neuro-oncology. Studies that investigated ASO therapy’s efficacy, CNS penetration, and safety profile were analyzed to assess its potential as a therapeutic intervention for high-grade gliomas. Results: ASOs present a promising avenue for enhancing targeted gene therapies in malignant gliomas. Their potent CNS penetration, in vivo durability, and efficient transduction offer advantages over conventional treatments. Preliminary in vivo and in vitro studies suggest ASOs as a viable adjuvant therapy for high-grade gliomas, warranting further exploration in clinical trials. Conclusions: ASOs hold significant promise as adjuvant therapy for high-grade gliomas, offering improved CNS penetration and durability compared with existing treatments. While preliminary studies are encouraging, additional research is needed to establish the safety and efficacy of ASO therapy in clinical settings. Further investigation and clinical trials are warranted to validate ASOs as a transformative approach in neuro-oncology.

## 1. Introduction

Glioblastoma (GBM) is the most common malignancy of the brain and central nervous system (CNS), making up 14.2% of all primary brain and CNS tumors, and 50.1% of malignant primary brain and CNS tumors [1]. The incidence rate of GBM is 3.26 per 100,000 population [1]. The mainstay treatment for GBM is surgical resection followed by adjuvant chemotherapy and radiation [2]. Despite recent advances in the management of these conditions, overall survival (OS) is still limited to approximately 16 months and individualized therapy efforts remain limited [3]. The underlying genetic mechanisms of disease play a significant role in the dismal outcomes of these patient populations, and there remains a serious need for the development of novel treatment strategies to target the genetic underpinnings of these conditions [4,5,6].

Antisense oligonucleotides (ASOs) are a form of targeted nucleic acid therapy capable of inhibiting the translation of specific genetic targets, thereby suppressing oncogenic genes or reactivating tumor-suppressive cellular programs [7,8]. ASOs are biochemically modified, single-stranded, deoxyribonucleotide molecules 15 to 25 nucleotides in length. They are created to be complementary sequences to the mRNA of a target gene, specifically suppressing the expression of that gene [9]. Composed of short single-stranded deoxynucleotides, ASOs can bind to mRNA via complementary base pairing and cause numerous effects, such as the inhibition of 5′ capping and polyadenylation, degradation via RNase H activity, steric hindrance of protein translation, and modulation of splicing products [10] (Figure 1).

By binding to its RNA target, ASOs form an RNA-DNA heteroduplex that acts as a substrate for RNase enzymes that break down the mRNA target in the cytoplasm [11]. ASOs can be chemically altered to increase the affinity for both its target and RNase activation. Additionally, ASO activity can be initiated soon after transcription. In the nucleus, oligonucleotides can bind to the site of addition of the 5′ cap or the poly-A tail, irreversibly blocking the active site for the enzymes guanyl transferase and poly-A polymerase, respectively [11]. Without these stabilizing additions, the transcriptional product is subjected to further degradation. ASOs can also induce alternative splicing mechanisms that alter the target transcript’s reading frame, induce exon skipping, and produce premature stop codons for shorter functional proteins [11]. Alternatively, the ASO can also induce exon inclusion by altering the structure of the spliceosome and preventing the removal of the exon [11]. 

The idea of ASO-mediated gene silencing has come as a potentially effective complement or alternative to traditional cancer treatments [12]. ASOs have been studied as potential therapeutics in other cancers, yet their role as novel treatment agents in the field of neuro-oncology is limited [10,13]. 

Gene therapy, immunotherapy, ASOs, small molecules inhibiting farnesyltransferase, tyrosine kinase, and matrix metalloproteinases are among the molecular pharmacotherapeutic approaches against GBM. The use of ASOs as a form of neuro-oncological therapy is an exciting and promising concept. This novel approach is predicated on an ASO sequence-specific binding to the target mRNA. The inhibition of gene translation is made feasible by this mechanism. In this study, we aimed to review the current advancements in the use of ASOs for neuro-oncology with a specific focus on GBM.

## 2. Chemical Modifications

Early models of ASOs showed decreased efficiency in vivo and high rates of decay, primarily due to the activity of the abundant exo- and endonucleases found in bodily fluids and within the cell. First-generation ASOs were also poorly tolerated due to their immunostimulatory properties. To combat these shortcomings, several chemical modifications have been made since the inception of ASO therapy to increase the half-life, circumvent nuclease activity and innate immune activation, and increase the potency [14]. Some literature shows that the unmodified phosphate backbone of oligonucleotides displays an increased affinity for nuclease degradation [4]. In contrast, phosphorothioate (PS) backbones, in which one of the non-bridging oxygens of the phosphate group is replaced with sulfur, substantially decrease the nuclease activity and enhance the target affinity [14]. This simple chemical substitution confers a great advantage to ASOs in vivo by facilitating the evasion of conventional nuclease activity. Furthermore, PS ASOs efficiently recruit RNase H to cleave their target mRNA and have increased affinity for binding to plasma proteins, which reduces the rate of clearance and enhances receptor-mediated uptake into target cells [15]. In contrast, phosphorodiamidate morpholino oligomers (PMOs) are DNA analog ASOs in which the phosphate backbone is completely replaced with a series of morpholine rings connected by phosphorodiamidate linkages [14]. This specific modification does not trigger RNase H-mediated cleavage and is common in ASOs that alter splicing products or inhibit protein production via steric hindrance due to the enhanced resistance to protease and nuclease activity [14]. However, in contrast to PS ASOs, which have a negatively charged backbone, PMOs are uncharged; this property increases their safety profile in vivo due to a reduction in off-target interactions with host proteins but contributes to a short serum half-life and poor cellular uptake compared with anionic ASOs. Productive uptake can be significantly enhanced by the addition of a targeting ligand or cell-penetrating peptide (CPP).

Apart from alterations that increase the half-life of ASO, several adjustments have been developed that lessen the degree of toxicity and enhance the stability. 2′-O-methyl/2′O-methoxyethyl RNA substitutions (2′-O-Me/2′-MOE), in which the 2′ hydroxyl (-OH) group of the ribose ring is conjugated to a methyl or methoxyethyl group, are common in contemporary ASOs with decreased off-target effects and drug-related side effects [14] (Figure 2). 

This substitution also increases the affinity of the ASO for its target and contributes to nuclease resistance. Since the 2′-MOE modification can only be performed in RNA moieties, pure 2′-MOE ASOs do not recruit RNase H, which specifically recognizes DNA-RNA complexes. Due to this limitation, ASOs are commonly constructed as chimeric “gapmer” oligonucleotides, in which a central PS-modified DNA “core” of at least 10 nucleotides is flanked by 2′-MOE-modified RNA “wings” at the 5′ and 3′ ends.

A less common chemical modification found in several ASOs contains a constrained nucleotide backbone characterized by a covalent bridge connecting the 2′ oxygen to the 4′ carbon of the ribose molecule [16]. This bridge forces a non-modifiable conformation onto the sugar backbone and works to increase the target affinity and avoid nuclease degradation. However, locked nucleic acids (LNAs) remain a relatively uncommon modification to ASOs since they have been linked to hepatocellular toxicities [16]. (S)-constrained ethyl (cEt) ASOs are a more recent version that also contains a methylene bridge that links the 2′-oxygen and 4′-carbon of the ribose but has been suggested to have reduced hepatotoxic potential. cEt-containing ASOs are currently being investigated in clinical trials but none have been FDA approved to date. Most contemporaneous oligonucleotide-based drugs contain a combination of these modifications to overcome the limitations that decreased the efficiency of first-generation ASOs. 

## 3. Delivery Methods

The blood–brain barrier is a significant obstacle for therapeutic options aimed at malignancies within the CNS, including ASOs. To circumvent this hurdle, several studies identified successful ways to deliver ASOs in vivo, providing necessary dosages while maintaining safety and efficacy profiles. Antibody conjugation to oligonucleotides facilitates diffusion across the BBB and cell membrane. Antisense strands bound to target antibodies are readily taken across the barrier and identify target cells [17]. Due to their small nature and immune ability, a minute number of antibodies can diffuse into tumor-rich areas. The oligonucleotide–antibody conjugation induces endocytosis and enhances permeability of the BBB and cellular membrane. In addition, the specific target antigen guides the ASO to the cells of interest and permeabilizes the membrane for its uptake, wherein the antisense strand is released from the antibody.

Select studies highlighted the use of ASO-derived nanoparticles to penetrate the BBB and enhance drug delivery. Cytosine–guanosine oligonucleotides bonded to apolipoprotein polymersomes were intravenously injected into a murine glioma model and evaluated for drug permeability [18]. The nanoparticle, formulated into a polycarbonate vesicle, allowed for intracellular reduction and uptake of the immunostimulatory oligonucleotide. Compared with free CpG-ODN, the nanoparticle showed significant increase in intracranial delivery. Similarly, the ASO enhanced the dendritic cell presentation and activation of anti-tumor chemokines and cytokines [18].

ASO modifications that contain folate-polyamidoamine dendrimers show a favorable increase in target suppression and cellular uptake [19]. The intrinsic positive charge on Folate-PAMAM provides a suitable surface for the stabilization of negatively charged oligonucleotides while shielding it from endonuclease activity and providing efficient migration into the nucleus. In vivo studies of ASOs conjugated to Folate-PAMAM showed a significant decrease in glioma size and proliferation [19]. Non-ablative radiation prior to ASO treatment remarkably increased the uptake and response in vivo [20,21]. This method temporarily disrupts the BBB and allows for the diffusion of targeted therapy into the tumor bed. Radiation enhances physiological endocytosis, leading to distribution within malignant cells [20]. In addition to resistance from the BBB, tumor microenvironments often oppose therapeutic avenues. Select studies manipulated the use of neural stem cells to accelerate the delivery of oligonucleotides to tumor-rich areas. Neural stem cells have favorable migration to malignant regions and maintain ASO integrity [22]. Using conventional cell-mediated exosome formation, extracellular vesicles containing ASO were successfully produced and trafficked into tumor-rich areas. The corresponding ASO was then released and significantly decreased the RNA translation and tumor size [22]. Similar studies were done that investigated the use of viral vectors for drug delivery and the results showed promising attributes, as ASOs conjugated to attenuated viral vectors enhanced the cellular uptake while reducing the protein expression in vitro [23]. 

Different delivery methods have been tried for ASOs. Nusinersen, an ASO that modulates the splicing of SMN2 RNA, was shown to exhibit profound disease-modifying effects in patients with spinal muscular atrophy, sparking renewed efforts to develop ASOs for other neurological diseases [24]. They present for the first time the atlas of ASO distribution and activity in the CNS of mice, rats, and non-human primates, which are species commonly used in preclinical therapeutic development. After central administration (intrathecal or intraventricular) of an ASO to rodents, researchers observed a widespread distribution throughout the CNS and a reduction in target RNA within neurons, oligodendrocytes, astrocytes, and microglia [24].

## 4. Proof-of-Concept Studies 

An array of in vivo/in vitro studies underscored ASOs’ potential as a therapeutic agent in neuro-oncology. Conceptual studies also divulged innovative ways to bypass entry into the blood–brain barrier (BBB) and increase the cellular uptake within the nucleus [9,10]. Furthermore, adjuvant ASO therapy was demonstrated to decrease the expression of known oncogenic sequences and to challenge the viability of cancer cells [6,7,8]. ASOs are being investigated in several clinical trials for their efficacy [25,26,27,28,29,30,31]. The outcomes of preclinical and/or clinical research using ASO therapy in cerebral gliomas are reported in this study. 

### 4.1. ASOs In Vitro

The ability of ASOs to decrease mRNA expression and protein translation in vitro has long been studied and affirmed. Select studies further confirmed its ability to target actionable mutations within glial cells and significantly decrease their expression compared with untreated controls [12,17,32,33,34,35,36]. In addition to altering gene expression at the transcriptional and translational levels, ASOs can decrease the expression of downstream targets, providing alternative avenues to target neurological malignancies [12,34]. A study by Murphy et al. demonstrated that comparable dosages of siRNA and ASO were enough to decrease protein expression, such as FGF, while ASO treatment simultaneously decreased cell proliferation [12,17,32,34]. Furthermore, ASO therapy selectively induces programmed cell death in tumor cells that harbor targeted mutations/deletions, while posing no threat to cells without the expression of the target sequence [12,33,37]. In addition to its selective capabilities, certain chemical modifications confer cellular specificity within the nervous system. ASOs composed of locked nucleic acids, in which there is an extra bridge locking the 2′ O to the 4′ carbon, create a favorable environment for neuronal cell entry [12]. Comparably, an ASO derived with a 2′-deoxy-2′-fluoro-beta-D-arabinonucleic acid (FANA) further increased the potency while avoiding endonuclease activity [17]. Arnold and colleagues designed a chimeric DNA-FANA ASO directed against downregulated in renal carcinoma (DRR)/FAM107A, which is a gene that is highly expressed in invasive glioblastoma stem cells (GSCs) and promotes GSC invasion. DDR/FAM107A-directed ASOs conjugated to the GSC antigens CD44 or EphA2 successfully decreased glial stem cell proliferation and self-renewal, which is a key indicator of malignancy in GBM [17]. These represent advantageous components of ASOs that favor their implementation within the field of neuro-oncology. 

Liu et al. utilized LNA ASOs to knockdown TP53 (p53) in U87 cells and discovered 99% knockdown efficiency as proof that ASOs can effectively reduce gene expression [38]. They then performed a CRISPRi screen to identify long noncoding RNAs that contributed to radiation resistance in glioma cells (lncRNA glioma radiation sensitizers (lncGRSs)). The authors evaluated the effectiveness of LNA ASOs against lncGRS-1 in glioma cell cultures generated from patients. A mean knockdown of 89% was achieved in patient-derived GBM SF10360 cells by two distinct ASOs against lncGRS-1, while a knockdown of 93% was achieved in patient-derived diffuse intrinsic pontine glioma (DIPG) SF8628 cells. Targeting lncGRS-1 with ASOs reduced tumor development and made glioma cells more susceptible to radiation treatment. These studies established a generalizable method to quickly find new therapeutic targets in the large noncoding genome and identify lncGRS-1 as a glioma-specific therapeutic target [38]. Jiang et al. further demonstrated that glioma-stem-cell-associated lncRNA suppression by either short hairpin RNAs (shRNAs) or ASOs reduces tumor cell proliferation and migration, as well as the capacity to generate xenograft tumors [39].

### 4.2. ASOs In Vivo

Engelhard et al. investigated the uptake and toxicity of urokinase-directed PS ASOs in rat carcinomatosis and rat brain tumor models, as well as their impacts on rat and human glioma cells in vitro [36]. Additionally, in vivo studies at dosages above anticipated therapeutic levels showed uptake into tumor cells without detectible toxicity. These results are encouraging for continued research into ASOs as a novel therapeutic approach for malignant glioma. Zhang et al. showed the benefit of this therapeutic approach in diffuse intrinsic pontine gliomas (DIPGs) by studying 2′-O-MOE-modified PS ASO gapmers, which direct the RNase H-mediated suppression of H3-3A mRNA [35]. In patient-derived neurospheres, they discovered a lead ASO that efficiently decreased H3-3A mRNA and H3.3K27M protein and restored global H3K27 trimethylation [35]. Two mouse models of DIPG were used to evaluate the lead ASO: an orthotopic xenograft using patient-derived cells and an immunocompetent mouse model using transduced mutant human H3-3A cDNA [35]. In both cases, ASO treatment restored the K27 trimethylation of histone H3 proteins; inhibited the growth of the tumor; promoted the differentiation of neural stem cells into astrocytes, neurons, and oligodendrocytes; and improved survival [35].

There are studies that demonstrated the use of ASO to inhibit the translocation of PKC isoforms to specific anchor proteins [40,41,42,43]. It was demonstrated that the PS-ASO ISIS 3521, which is a 20-mer oligonucleotide that suppresses PKCalpha expression, significantly reduced tumor invasion and proliferation in a rat model [33]. However, ISIS 3521 did not show a significant benefit in a phase II trial of untreated colorectal cancer, possibly due to the limited tolerability and half-life of PS-only ASOs compared with newer antisense generations [44,45]. The expression of the cyclin-dependent kinase (CDK) inhibitor p21WAF1/CIP1 was shown to increase as the PKC isoform activity levels decreased, and its elimination by a specific plasmid-expressed antisense cDNA inhibited PKC-induced cell cycle progression [46]. In addition, a strong correlation was shown between the level of PKC overexpression and the malignant behavior of C6 rat glioma cells [47]. 

Nagato et al. used PMO-ASO to target the extracellular matrix protein laminin [48]. The extracellular matrix’s structural component known as the laminin family plays an essential role in fostering the mobility of infiltrating tumor cells. Laminin 4 expression was downregulated for up to 3 days after the glioma cells were exposed to an ASO for the laminin 4 chain (AS-Ln-4) in culture [48]. The adhesion and migration of glioma cells were markedly reduced by AS-Ln-4 [48]. Additionally, invasiveness was dramatically decreased in cells transfected with AS-Ln-4 for 4 h, as opposed to cells transfected with the sense oligonucleotide for the same amount of time (S-Ln-4) [48]. In an orthotopic glioma model, AS-Ln-4-transfected cells failed to penetrate nearby normal brain tissues when glioma spheroids were implanted into rat brain slices [48].

### 4.3. ASOs as Adjuvant Therapy

Apart from their ability to diminish the expression of oncogenic sequences, ASOs work in tandem with current treatment methods to further decrease the viability of cancer cells. Oligonucleotide therapy seemingly diminished aberrantly expressed genes while making cells more sensitized to treatment with chemoradiation [23,49,50]. Belenkov et al. demonstrated that a 2′-MOE/PS ASO gapmer that targets human Ku86 messenger RNA, M059K, sensitizes malignant glioma cells to ionizing radiation, Bleomycin, and Etoposide [51]. According to their findings, Ku86 antisense medication in combination with radiotherapy and/or radio mimetics may be a helpful treatment to overcome radioresistance in several malignancies [51]. Furthermore, ASOs can be engineered to specifically target actionable mutations that confer chemotherapy resistance, such as MGMT and WISP1 [52,53]. Apart from significantly decreasing the corresponding gene expression, in vivo models of glial cells showed a favorable statistically significant response to chemoradiation when treated in tandem with ASOs [20,23,50]. In addition to increasing the susceptibility, ASO therapy has been implicated in improving the immune response within immunosuppressive areas. Tumor models treated with oligonucleotides show the differential expression of common immune markers indicating heightened immune response [12,22]. 

### 4.4. Clinical Trials

The tolerability and effectiveness of ASOs are currently being tested in several clinical trials. Though systemic oligonucleotide therapy is often well tolerated, adverse effects, such as thrombocytopenia, hypotension, fever, and asthenia, are dictated by the chosen chemical modifications and are dose-dependent [54,55]. 

A phase II trial in 2005 investigated the efficiency of the previously mentioned ISIS 3521/aprinocarsen PS-modified oligonucleotide against Protein Kinase C (PKC) in patients with GBM (Table 1). The overexpression of PKC-alpha has been associated with tumor proliferation and progression; therefore, an ASO that mediated RNase H degradation of PKC was continuously infused over 21 days in this cohort [25]. The preliminary results reported that patients treated intravenously with this drug experienced increased intracranial pressure and displayed rapid tumor progression and neural deficit, possibly due to the pro-inflammatory nature of first-generation PS-only ASOs [25]. Plasma ASO concentrations were also highly variable between patients, perhaps due to the limited half-life of the ASO or the chosen route of delivery. The study found no clinical benefit for this therapeutic method despite initial promising in vitro and in vivo outcomes. 

A phase I/II trial that investigated a PS-only ASO (AP 12009/trabedersen) targeted toward Transforming Growth Factor-Beta (TGFB) in GBM and anaplastic astrocytoma (AA) found a statistically significant increase in the overall survival of patients treated intratumorally with this drug [27]. TGFB has been implicated in fostering an immunosuppressive environment within tumor-rich areas, as well as an upregulation of cell signaling pathways leading to increased growth and malignancy [26,27]. Rich et al. showed that whereas malignant glioma cell lines may produce significant mediators of the pro-tumorigenic effects of TGFB but are not growth suppressed, TGFB treatment caused cell cycle arrest in astrocytes, which are the suspected cells of origin of astrocytomas [56]. TGFB signaling can be interrupted by ASOs [57]. In a preclinical study using 9L glioma in Fischer 944 rats, intracranial TGFB-2 ASO administration and vaccination with exposed glioma cells together not only inhibited TGFB protein production in vivo as expected but also significantly increased the survival times in comparison with either the vaccine alone or no therapy [58]. Patients in the study were treated intratumorally using the convection-enhanced delivery of Trabedersen, which is a synthetic antisense phosphorothioate oligodeoxynucleotide that induces mRNA degradation of the human TGFB-2 gene [27]. There was a significant difference in the 14-month tumor rate between participants treated with 10 μM of Trabedersen vs. those given conventional chemotherapeutics in those with AA [26,27,30,31]. No significant trend was established in participants with GBM. In addition, the study reported less drug toxicity and increased time to recurrence in those treated with the ASO therapy vs. those given chemotherapeutic agents [26,27,30,31]. 

The use of immunostimulatory ASOs in new-onset GBM were studied in a phase II trial. Unmethylated CpG-ODN were shown to induce the activation of Toll-like receptor 9 (TLR9), enhancing the antigen presentation and improving the immune profile of several cancers, including GBM [59]. As such, the randomized study assessed the role of injection of CpG-28 into the tumor bed post-resection on the activation of the immune response, as well as survival. The two-year OS was 31% in the treatment group compared with 26% in the placebo group, with no difference in the median PFS. The study found no impact on the survival between those expressing high vs. low levels of TLR9. The treatment toxicity was limited to fever and post-operative hematoma [60]. 

A pilot immunotherapy trial analyzed the use of an ASO, IMV-001, against Insulin-like Growth Factor Type 1 (IGF-1) in GBM and the ability to mount an anti-tumor immune response. This receptor regulates key processes, including transformation, cell survival, and the inhibition of apoptosis [29]. Malignant cells were intraoperatively harvested from patients with GBM, treated with ASOs, and re-implanted into the abdomen within a biodiffusion chamber [28,61]. Western blot analysis confirmed a decreased IGF-1 expression after ASO treatment. The results of a phase Ib trial showed that patients treated with ASO had a median PFS of 9.8 months and an OS of 17.3 months. Furthermore, the study reported that treatment with IMV-001 led to a heightened immune response, which could play a role in controlled cell death [28,61]. Overall, this study reported a high safety profile with minimal complications limited to hematomas and wound complications at the implantation site [28,62]. A multicenter phase IIb trial evaluating the effect of IMV-001 on progression-free survival compared with a placebo is ongoing [62,63].

Additionally, preliminary trials studied the use of oligonucleotides in pediatric gliomas, including medulloblastomas, ependymomas, and diffuse pontine gliomas [64]. Imetelstat, which is an ASO that targets telemorase activation, was administered intravenously pre- and post-surgical resection [65]. The primary aim of the study was to evaluate telomerase inhibition, which found knockdown of the enzyme to persist for 8 days, returning to baseline 21 days post-injection. Patients enrolled in the study demonstrated various toxicities that led to a premature arrest of the trial, including thrombocytopenia, lymphopenia, neutropenia, and intratumoral hemorrhage [65]. 

## 5. Limitations

Despite recent advances in the development of oligonucleotide therapy, many aspects pose a challenge to their implementation, with the primary being blood–brain barrier penetration. ASOs have a high molecular weight and a significant negative charge, which prevents them from passively entering cellular membranes. When given intravenously, only a fraction of the dose makes it into the cerebral hemispheres, requiring higher concentrations to reach a therapeutic level in the CNS [14]. With increased drug concentrations comes the increased likelihood of off-target effects and drug-induced toxicity. Further work aimed at enhancing the specificity of ASOs for their respective targets may minimize the severity of medication side effects and allow for a broader and more efficacious application of adjuvant ASO therapy. Potential alternatives to intravenous infusion include direct intraoperative injection to the tumor bed or post-operative intrathecal injection. Furthermore, the systemic circulation of endonucleases and exonucleases have a significant propensity to degrade exposed ASOs. Intrathecal injection, which is a well-established and proven method for the delivery of FDA-approved ASOs, has demonstrated both efficacy and excellent tolerability in clinical settings [66]. This route of administration involves the direct injection of ASOs into the cerebrospinal fluid, allowing them to access the central nervous system with remarkable success. Nusinersen, which is an antisense oligonucleotide employed in spinal muscular atrophy, exemplifies the profound therapeutic efficacy achievable through the intrathecal administration of antisense oligonucleotides [66]. Notably, the majority of ASOs designed to target gliomas currently in clinical use are of the first generation. This progress highlights the importance of ongoing research and development in the field of ASOs to continually refine and optimize their therapeutic potential. Once inside the BBB, the oligonucleotides must be taken up by the cell within endosomes and trafficked into the nucleus. Cellular uptake and migration into the nucleus are key steps in ASO targeting that pose a barrier to its efficacy. Endosomal modifications of ASOs decrease their targeted effect and hinder their ability to reach the nucleus [14]; however, select studies identified ways to increase nuclear uptake while avoiding cellular modifications. Nonetheless, increasing nuclear uptake within neural cells warrants further research, as it could enhance the potency of ASOs. 

## 6. Conclusions

Antisense oligonucleotides target expressed sequences within the genome and downregulate their expression, providing a novel treatment strategy for high-grade gliomas due to the aberrant genetic code underlying these conditions. Despite promising in vitro and in vivo data, limited studies have been done on effective delivery methods and regulation of off-target effects. Clinical trials focused on addressing these limitations are needed to optimize the adjuvant use of ASOs, as they offer promising new insights into genetics-based treatments for these conditions. 

## Figures and Tables

**Figure 1 cancers-16-01944-f001:**
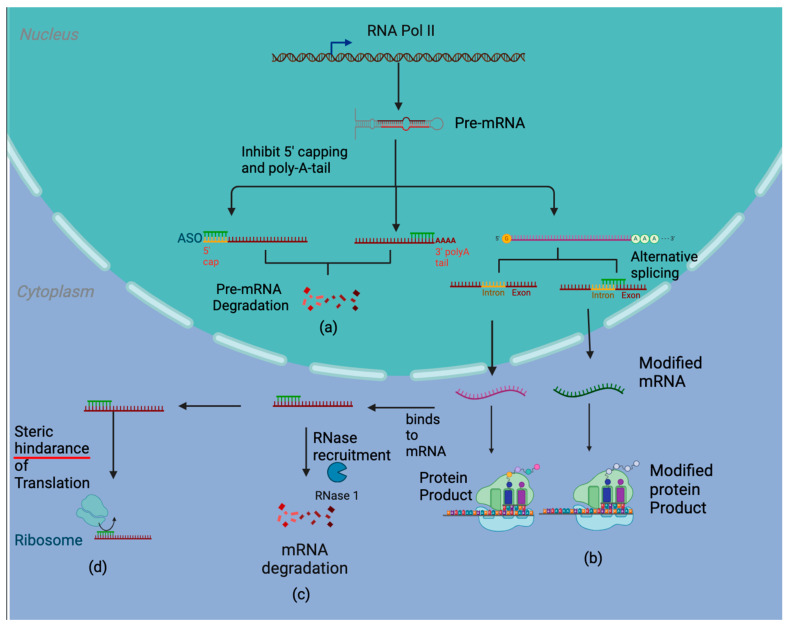
ASOs can decrease the formation of protein products by blocking the addition of the 5′ cap or the poly-A tail or by causing alternative splicing to recently transcribed mRNA. ASOs can similarly block translation by inhibiting the binding of ribosomes to mRNA or by recruiting RNase and causing the degradation of RNA.

**Figure 2 cancers-16-01944-f002:**
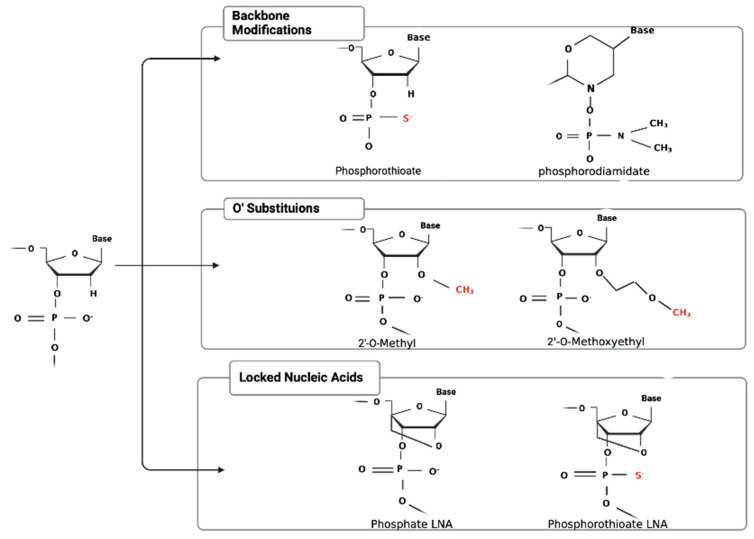
Chemical modifications of ASO include backbone remodeling, 2O’ substitutions, and locked nucleic acids. Chemical modifications of ASO work to stabilize the product and increase the efficacy.

**Table 1 cancers-16-01944-t001:** Summary of Selected Antisense Oligonucleotide (ASO) Drugs, Their Modifications, Mechanisms, Routes of Administration, Safety Profiles, and Clinical Outcomes.

Drug Name	Chemical Modifications	Mechanism	Route of Administration	Safety Profile	Outcomes
Aprinocarsen (ISIS 3521)	Phosphorothioate backbone	RNase H degradation	Intravenous infusion	Increased intracranial pressure, tumor progression, and neural deficit	No clinical benefit
Trabedersen (AP12009)	Phosphorothioate backbone	mRNA degradation	Intratumoral injection	Fever, headache, and GI discomfort	Increased time to recurrence
CpG-ODN	Cytosine triphosphate linked to guanine triphosphate	Activation of TLR-9	Intratumoral injection	Fever and post-op hematoma	Increased OS
IMV-001	Single-stranded 18-mer oligonucleotide	Steric hindrance of translation	Abdominal bio-diffusion chamber	Hematomas and wound complications	Increased PFS and OS
Imetelstat	Thiophosphoramidate oligonucleotide	Competitive inhibition of Telomerase Enzyme	Intravenous injection	Thrombocytopenia, lymphopenia, neutropenia, and intratumoral hemorrhage	Persistent knockdown of telomerase

## Data Availability

Data are contained within the article.

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
