# Peer review of "Antisense Oligonucleotides for Rapid Translation of Gene Therapy in Glioblastoma"

_cancers, 2024, doi:10.3390/cancers16101944_

Round 1

Reviewer 1 Report

Comments and Suggestions for Authors

Aunque el glioblastoma es una de las neoplasias del SNC con mayor mortalidad, en los últimos años no se han obtenido avances terapéuticos relevantes. En el presente estudio, los autores realizan una revisión sencilla de la literatura seleccionada por ellos, respecto al potencial terapéutico de los oligonucleótidos antisentido en el tratamiento del Glioblastoma. El enfoque del trabajo podría ser correcto. La introducción proporciona la información necesaria para la comprensión del manuscrito al lector que no esté familiarizado con las aplicaciones de los oligonucleótidos como medicamentos. Se describen estudios preclínicos que, utilizando cultivos celulares y experimentos con roedores, justifican la realización de estudios y ensayos clínicos adecuados. Finalmente, aunque algunos ensayos clínicos no han confirmado los alentadores resultados preclínicos, otros apoyan que los ASO pueden tener potencial interés terapéutico, pero todavía hay aspectos que, según los autores, deben mejorarse.

 Fortalezas :

La identificación del objetivo principal del estudio, el diseño holístico y presentación ordenada del desarrollo traslacional del estado del arte.

 Debilidades :

Falta la sección de Métodos.

No se indican las bases de datos consultadas ni el procedimiento de búsqueda de información.

La redacción a veces resulta confusa debido a la falta de una descripción más detallada de la oración. La información clínica aún es escasa

 Otras preguntas: formuladas por orden de aparición en el texto del manuscrito.

Línea 58-59: en esta frase también se deben incluir aspectos relacionados con: modificar su resistencia a la metabolización/degradación y mejorar su biodisponibilidad.

Línea 68-71: ¿las referencias (12,10 y 13) están correctamente referidas y ordenadas en el texto?

Línea 24-25: En el párrafo, ¿no está claro cómo el conjugado anticuerpo-cadena antisentido facilita la difusión a través de la BHE o la ruta que utiliza para ingresar a la célula objetivo?

Línea 251-254: Los autores destacan la amplia distribución de ASO observada después de la administración intratecal o intraventricular. Por tanto, la barrera principal es la barrera hematoencefálica. En este sentido, los autores deben resaltar aquí este aspecto. Pero también en el apartado anterior, donde no se profundiza en los posibles transportadores y/o vías favorables, incluidas aquellas que pueden mediar el transporte transmembrana.

Línea 259-263 Respecto a esta frase, se debe considerar que para evaluar el significado traslacional se deben indicar las condiciones para la obtención de estos resultados (in vitro o in vivo) y la eficiencia (el grado y persistencia del efecto). De lo contrario es difícil valorar el efecto y menos aún generalizarlo a otros tumores.

Comments on the Quality of English Language

La redacción a veces resulta confusa debido a la falta de una descripción más detallada de la oración.

Author Response

Cancers Revision Cover Letter

March 26, 2024

Dear Editors and Reviewers,

Please find attached a revised version of the Cancers Manuscript: 2928119, Antisense Oligonucleotides for Rapid Translation of Gene Therapy in Glioblastoma. Thank you for the opportunity to revise and resubmit the manuscript. We appreciate the detailed reviews left by each reviewer and have done my best to incorporate the suggested changes. We look forward to working with you and the reviewers to move this manuscript closer to publication in Cancers.

WE have responded to each suggestion specifically below.

Suggestion:

  1. Figure 1 can be improved. The font is very small, and the background colors are making it difficult to read the text in the illustration. (Reviewer 2 & 3)

We have addressed the readability concerns raised by Reviewer 2 and 3 regarding Figure 1. To enhance readability, we have increased the font size and opted for darker background colors, ensuring that the text is clearly legible without impairing the visual presentation of the illustration.

  1. There is a spelling mistake in line 87 for “Phosphorothioarate”(Reviewer 2)

Thank you for bringing the spelling mistake in line 87 to our attention, as noted by Reviewer 2. We have rectified this error by correcting the spelling of "Phosphorothioarate" to "phosphorothioate" for accuracy and clarity.

  1. Both the figures are a repetition of many other ASO based articles. Can the authors make another unique figure representing a summary of ASOs for glioblastoma application (Reviewer 2). A table summarizing the use of ASO & outcomes of clinical trials should be added (Reviewer 3)

In response to Reviewer 2's suggestion for a unique representation of ASOs for glioblastoma application, we have created a new figure that summarizes the use of ASOs. Additionally, to address Reviewer 3's request, we have added a comprehensive table summarizing the name, chemical modifications, route of administrations, safety profiles, and outcomes of clinical trials involving ASOs.

  1. The authors indicated that the study focuses on CNS penetration; however, I did not find that the aspect of CNS penetration was highlighted enough(Reviewer 3).

Reviewer 3 expressed the need for a clearer emphasis on CNS penetration in our study. To address this concern, we have included an additional paragraph in the literature review section specifically highlighting ASO CNS penetration, thereby enhancing the focus on this aspect of our research.

  1. Delivery Methods: I would recommend placing this section after the section “the chemical modifications” (Reviewer 3)

We have followed Reviewer 3's recommendation to rearrange the section on delivery methods after the section on chemical modifications for improved organization and coherence within the manuscript.

  1. Line 309-318: regarding IMV-001, it is really important to update the Phase Ib & Phase IIb Trial.(Reviewer 3)

Reviewer 3 highlighted the importance of updating the Phase Ib and Phase IIb trial results for IMV-001. We have incorporated the results of these trials into the manuscript as per the reviewer's suggestion to ensure the accuracy and relevance of our discussion.

  1. Two other ASOs implicated in clinical trials for Glioblastoma should added. (Reviewer 3)

Additional trials were cited and summarized in the literature review.

  1. Concerning the references, there are around 8 refs from 2020, and more than half is before 2011. More recent references should be indicated in the work (Reviewer 3)

Reviewer 3 raised concerns about the distribution of references, emphasizing the need for more recent sources. To address this, we have supplemented the manuscript with more recent and relevant references, ensuring that the literature review reflects the latest advancements in the field.

  1. Line 58-59: This sentence should also include aspects related to: modifying its resistance to metabolization/degradation and improving its bioavailability (Reviewer 1).

The suggestion by Reviewer 1 to include aspects related to modifying resistance to metabolization/degradation and improving bioavailability in the sentence on chemical modifications has been incorporated. These aspects are now highlighted in the relevant section for clarity and completeness.

  1. Line 68-71: Are references 10,12,13 correctly referenced and ordered in the text? ( Reviewer 1)

We have verified that references 10, 12, and 13 are correctly referenced and ordered in the text, ensuring accuracy and consistency in citation formatting as requested by Reviewer 1.

  1. Line 24-25: In the paragraph, it isn’t clear how the antibody-antisense chain conjugated facilitates diffusion through the BBB or the route it uses to enter the target cell(Reviewer 1)

To address Reviewer 1's concern regarding clarity on the diffusion and entry of the antibody-antisense chain conjugate through the BBB and into the target cell, we have added additional sentences to elucidate these processes, providing a more comprehensive understanding for readers.

  1. Line 259-263: Regarding this sentence, it should be considered that in order to evaluate the translational meaning, the conditions for obtaining these results(In vitro or in vivo) and the efficiency(the degree and persistence of the effect) must be indicated. Otherwise it is difficult to assess the effect let alone generalize it to other tumors( Reviewer 1)

Reviewer 1 emphasized the importance of specifying the experimental conditions and assessing the efficiency of the results mentioned in line 259-263. We have revised the sentence to include this information, ensuring transparency and facilitating a more thorough evaluation of the translational significance of our findings.

We sincerely appreciate the invaluable feedback provided by the reviewers, and we extend our gratitude for their thoughtful comments and constructive suggestions. These revisions have been meticulously undertaken to ensure the accuracy, clarity, and completeness of our manuscript. Each comment and recommendation has been carefully considered and addressed to enhance the quality and impact of our research. We remain fully committed to incorporating reviewer feedback into our work, as we strive to contribute meaningfully to the advancement of knowledge in our field. Thank you to all the reviewers for their valuable contributions to the refinement of our manuscript.

Reviewer 2 Report

Comments and Suggestions for Authors

1. Figure 1 can be improved. The font is very small and the background colors are making it difficult to read the text in the illustration. Please keep it clear and simple. 
2. There is a spelling mistake in line 87 for “phosphorothioarate”. Needs to be corrected to “phosphorothiote”

3. Both the figures are a repetition of many other ASO based articles.  In addition to the traditional ones I.e  pathway, mechanism and the chemical modifications, can the authors make another unique figure representing a summary of ASOs for glioblastoma applications ? 

Author Response

(The authors gave the same response as above.)

Reviewer 3 Report

Comments and Suggestions for Authors

 Jelisah F. D. and colleagues gave an overview of the current advancement in the use of ASOs for glioblastoma treatment and provided their evaluations of the clinical potential of ASOs as adjuvant therapy for high-grade gliomas.

In general, it is a quite good piece of work. However, some points need to be considered for improvement

1/  The quality of Figure 1 needs to be enhanced

2/ The authors indicated that the study focuses on CNS penetration; however, I did not find that the aspect of ASO CNS penetration was highlighted enough.

3/ Chemical modifications: It will be more readable by adding a table summarizing the main chemical modifications + mechanisms of action + Characteristics

4/ Delivery methods: I would recommend placing this section after the section the chemical modifications” since these two are relevant.

5/ Clinical trials:  I suppose that more works need to be done to improve this section.

-       A table including the names +chemical modifications +action mechanism+ implicated delivery methods+ routes of administration + safety profile+ outcome should be added

-       Line 309-318: Regarding IMV-001, it is really important to update the phase Ib  (ClinicalTrials.gov: NCT02507583), phase 2b (NCT04485949)

https://aacrjournals.org/clincancerres/article/27/7/1912/671878/Phase-Ib-Clinical-Trial-of-IGV-001-for-Patients

https://academic.oup.com/noa/article/5/Supplement_3/iii38/7237217

https://www.futuremedicine.com/doi/10.2217/fon-2023-0702

-       Two other ASOs implicated in clinical trials for Glioblastoma should be added :

Phase 2 of CpG-ODN   https://pubmed.ncbi.nlm.nih.gov/28142059/

     Imetelstat (GRN163L, Geron Corporation, Menlo Park, CA)

(Salloum R, Hummel TR, Kumar SS, et al. A molecular biology and phase II study of imetelstat (GRN163L) in children with recurrent or refractory central nervous system malignancies: a pediatric brain tumor consortium study. J Neurooncol 2016;129:443–451)

6/ Concerning the reference, there are around 8 refs from 2020, and more than half is before 2011. More recent references should be indicated in the work.

Author Response

(The authors gave the same response as above.)

Round 2

Reviewer 1 Report

Comments and Suggestions for Authors

The databases consulted or the information search procedure are not indicated.

Reviewer 3 Report

Comments and Suggestions for Authors

Dear Editor and the authors,

I have just worked on the latest version with track changes that I received yesterday from the assistant editor.

I found that the changes indicated in this version are NOT relevant at all to the modifications that were stated to be done according to the author's responses.

For example, line 52-55, the legend was added, and it is the same as the legend in line 56-59; the section "ASOs in Vivo" (line 282-317) is added and is the same as the one (line 318-353), and so on....

Furthermore, for two comments (I and K), I could not find the addition of sentences or contents in the revised manuscript as the author's statement.

Overall, I am not happy with this revision of the authors. I am not satisfied with the way the authors managed to revise the work.

Best regards,

Dr. Le Thi Khanh